# The Effects of Chronological Age on the Chondrogenic Potential of Mesenchymal Stromal Cells: A Systematic Review

**DOI:** 10.3390/ijms242015494

**Published:** 2023-10-23

**Authors:** Antonia Vogt, Konstantinos Kapetanos, Neophytos Christodoulou, Dimitrios Asimakopoulos, Mark A. Birch, Andrew W. McCaskie, Wasim Khan

**Affiliations:** 1Division of Trauma & Orthopaedic Surgery, Addenbrooke’s Hospital, University of Cambridge, Cambridge CB2 0QQ, UK; av591@cam.ac.uk (A.V.);; 2School of Clinical Medicine, University of Cambridge, Cambridge CB2 2SP, UK

**Keywords:** MSCs, chronological age, chondrogenic differentiation

## Abstract

Tissue engineering and cell therapy for regenerative medicine have great potential to treat chronic disorders. In musculoskeletal disorders, mesenchymal stromal cells (MSCs) have been identified as a relevant cell type in cell and regenerative strategies due to their multi-lineage potential, although this is likely to be a result of their trophic and immunomodulatory effects on other cells. This PRISMA systematic review aims to assess whether the age of the patient influences the chondrogenic potential of MSCs in regenerative therapy. We identified a total of 3027 studies after performing a search of four databases, including Cochrane, Web of Science, Medline, and PubMed. After applying inclusion and exclusion criteria, a total of 14 papers were identified that were reviewed, assessed, and reported. Cell surface characterization and proliferation, as well as the osteogenic, adipogenic, and chondrogenic differentiation, were investigated as part of the analysis of these studies. Most included studies suggest a clear link between aged donor MSCs and diminished clonogenic and proliferative potential. Our study reveals a heterogeneous and conflicting range of outcomes concerning the chondrogenic, osteogenic, and adipogenic potential of MSCs in relation to age. Further investigations on the in vitro effects of chronological age on the chondrogenic potential of MSCs should follow the outcomes of this systematic review, shedding more light on this complex relationship.

## 1. Introduction

Mesenchymal stromal cells (MSCs) have been the topic of much interest with regard to their potential and much of current research has intertwined varying definitions for what these specifically represent. The current established definition for these cells suggests that MSCs are whole-tissue-specific cells that can switch morphology into other cell types. Some of this has been clarified by seminal work from Caplan et al. [1], which has shown that the majority of MSCs are not stem cells. MSCs are now believed to secrete immunomodulatory and trophic cells that alter the biology of the region in which it is active [1].

MSCs were initially derived from the bone marrow [2], but the complexities and inconvenience associated with obtaining MSCs from this source [3] led to the development of protocols for the isolation of MSCs from pericytes and adventitial progenitor cells from other tissues in the body [4], including adipose tissue, peripheral blood, lungs, dental tissues (dental pulp stem cells (DPSCs), stem cells from human exfoliated deciduous teeth (SHED)) [5,6,7] and neonatal-birth-associated tissues (e.g., placenta, umbilical cord and cord blood) [8].

MSCs can differentiate into other mesoderm-derived specialised cells such as osteoblasts, chondrocytes, adipocytes, tenocytes, and myocytes [9,10,11], and are also capable of differentiating into non-mesoderm-derived cells such as glial cells and neural cells [12,13,14,15,16,17,18]. Morphologically, MSCs can be characterised by a spindle-like shape in the undifferentiated state [19]. These cells can also be characterised by the cell surface markers [20] they express or lack: MSCs typically express CD73, CD90, CD105 [21], cell adhesion molecules (e.g., CD54/ICAM–1 or CD106/VCAM–1) [22,23] and some cytokine receptors (e.g., IL–1R and TNF–aR). MSCs are also typically negative for the cell surface markers CD11b, CD14, CD19, CD34, CD45, CD79a HLA–DR and vWF [24,25]. A precise MSC characterization profile is difficult to establish due to the heterogeneity of MSCs. The International Society for Cellular Therapy (ISCT) proposed the minimum criteria that characterize cells as MSCs [26].

MSCs have an important potential therapeutic use in human musculoskeletal disease. A key driver for research in MSCs is biological therapies for osteoarthritis (OA). OA is the most common articular cartilage degenerative disorder affecting different joints of the body, albeit mainly affecting the joints in the knee, hip, hands, and spine [27]. Characterised by articular pain, OA leads to impaired joint function and progressive disability [28]. Due to the lack of disease-modifying treatment for OA, current strategies are limited to analgesic control and lifestyle modification [29]. Past studies have shown that MSCs can differentiate into the main components of articular cartilage: (1) proteoglycans, providing compressive stiffness to the cartilage; and (2) type II collagen, contributing to tensile strength and resilience of the cartilage [30]. As with many pathologies, advancing age is associated with OA development; age in itself is not a pathological process but represents the accumulation of changes that contribute to disease. With over 75% of OA sufferers being those over 65 [31], the quality and integrity of MSCs in treating older patients have been questioned and studied.

Several properties of MSCs make them putative treatment options in diseases characterised by degeneration. These include (1) selective migration and homing to the inflammatory microenvironment [11,32]; (2) secretion of trophic factors, such as growth factors, cytokines, morphogens, anti-apoptotic factors, and exosomes [33,34,35]; and (3) immunoregulatory responses to antibody production, T cell activation and cytokine secretion by NK cells [9,36]. These properties make MSCs behave in a non-stem-cell-like fashion. Due to the poor regenerative potential of articular cartilage [37], the aforementioned properties of MSCs alongside their chondrogenic potential could make them useful in orthopaedic applications through their immunomodulatory effects on other cells as well as immune cells.

Mesenchymal progenitors give rise to chondrocytes in vitro that have the potential to lead to cartilage development. Chondrocytes are metabolically active cells that differentiate and proliferate during development and are the predominant cells in healthy cartilage [38]. However, there is no detection of cell proliferation in adult cartilage. Only 1–5% of cartilage is occupied by the chondrocytes in adults; the rest is the extracellular matrix [39]. A large volume of proteoglycans, collagen, glycoproteins and hyaluronan, also known as extracellular matrix compounds, are turned over and synthesized by chondrocytes [40]. The mechanical and chemical environment of chondrocytes are factors that affect their metabolic activity [41]. Although skeletal development can result from the proliferation of chondrocytes in vivo, it can also occur in vitro [42]. The formation of chondrocytes, resulting in cartilage development, is called chondrogenic differentiation and takes 28 days to form in vitro [43].

The osteogenic and adipogenic differentiation process of MSC in vitro lasts almost three weeks. Lineage commitment and maturation are the two main stages of these processes [44]. The first differentiation from bone marrow stem cells, described by Friedenstein et al. in the late 1960s, was the formation of osteoblasts. This differentiation was identified as osteogenic. The formation of fat cells—adipocytes derived from stem cells—is called adipogenesis, where preadipocytes differentiate into mature adipocytes. There are two cell populations adipocytes can arise from: either from bone marrow progenitor cells (migration to adipose tissue) or from preadipocytes (adipose-tissue-resident) [45].

MSCs, when cultured on fibronectin and bFGF-coated wells, undergo neurogenesis, differentiating into neuronal cells. After 14 days, they display characteristics of both neurons and glial cells, confirmed by positive markers NF200 (68.9%), GFAP (15.4%), and Gal-C (12.3%). This successful differentiation process holds potential for tissue regeneration in nervous system disorders and has been a key focus in clinical trials [5,46,47].

Advancing age is associated with declining MSC function, including reduced proliferation and differentiation potential, enhanced cellular apoptosis, and reduced wound-healing properties [48,49,50]. Epigenetic and genetic mechanisms have been shown to underlie these senescence-related changes [51]. Despite these findings, studies on the effect of ageing on the chondrogenic potential of MSCs have yielded inconsistent results. It is worth noting that the papers included in these studies are in their early stages, making it unlikely for them to undergo senescence. Additionally, the culture conditions remained consistent within each paper, indicating that any observed changes in age-related characteristics are not attributed to senescence [52,53].

This systematic review, therefore, aims to summarise the evidence and evaluate the methodological quality of studies that have examined the effect of ageing on the MSC chondrogenic potential. In addition, this systematic review assesses whether the chronological age of patients affected the proliferative and chondrogenic capacity for differentiation with mesenchymal stromal cells. Finally, this systematic review reports the available literature and studies that are present and amalgamates these together to formulate a review of these here.

## 2. Methods

### 2.1. Search Strategy

A systematic review of the literature was performed according to the Preferred Reporting Items for Systematic Reviews (PRISMA) guidelines [54]. A search of the literature was carried out exploring four databases: Cochrane, PubMed, Medline, and Web of Science. These were performed in the last week of July and the first week of August. For our search strategy, we used the following search terms: “age” or “aging” and “mesenchymal stem cells” or “mesenchymal stem cell” or “mesenchymal stromal cells” or “mesenchymal stromal cell” and “cell surface characterisation” or “cell surface” or “differentiation potential” or “differentiation” and “in vitro”.

In total, 23 studies were extracted from Cochrane (1946—last week of July 2023), 992 from PubMed (1996—last week of July 2023), 703 from Medline (1946—first week of August 2023), and 1309 from Web of Science (1900—last week of July 2023).

Overall, 966 papers were identified as duplicates and removed. Once the inclusion and exclusion criteria were applied and papers were screened based on abstract and title, 1927 papers were excluded. A total number of 144 studies were assessed based on the full text and excluded based on the exclusion criteria listed below. Following this process, which is explained in detail in Figure 1, we identified a final number of 14 papers, which we used for data extraction.

The search was conducted by A.V., D.A., K.K., and N.C., and two authors (W.K. and A.V.) independently screened titled abstracts. In cases of disagreement, papers were included for full review.

The study was registered on the PROSPERO database with the registration number (459279).

### 2.2. Inclusion Criteria

In vitro studies involving adult human subjects;Studies with a reference to subjects’ age;Studies looking at MSCs and the source of extraction of cells specified;Studies that refer to chondrogenic differentiation;English language.

### 2.3. Exclusion Criteria

Duplicate studies;Those not in the English language;Non-human studies;Studies using samples from patients with systemic diseases;Any paper other than research papers was excluded;Studies looking at non-mesenchymal cells, e.g., embryonic, umbilical cord, and periodontal MSCs.

### 2.4. Data Extraction

An Excel spreadsheet was used to present the data that were extracted from each study. The data extracted are presented in four tables listing the reference of the papers, a brief description of the study, subjects’ number and chronological age, source of the MSCs, culture conditions, proliferation analysis, MSC cell surface characterisation, and chondrogenic, adipogenic and osteogenic differentiation.

### 2.5. Quality Assessment

A quality check for each paper was carried out using a modified version of the “OHAT risk of bias rating tool for Human and Animal studies” from the Office of Health Assessment and Translation (OHAT tool) [55]. Any differences in the results were solved by discussion.

## 3. Results

### 3.1. General Characteristics of the Papers

In this systematic review, 14 studies were included, as presented in Figure 1. The earliest study that is included that met the inclusion and exclusion criteria is from 2007 [56] and the latest 2019. For most papers, MSCs were isolated from different tissues of healthy or osteoarthritic patients, and the proliferation and chondrogenic differentiation of MSCs were investigated. Moreover, some papers presented data for adipogenic and osteogenic differentiation. The samples used for each paper varied between 6 and 260. Seven papers compared the effects of age between three different age groups, three papers presented data for two age groups, three for one group (younger and older) and only one had four different age groups. Where possible, we excluded whole groups or data from people under 18 years old. Bone MSCs were isolated for seven papers. More specifically, bone marrow (BM) MSCs were isolated mainly from the iliac crest and tibia. Furthermore, MSCs of adipose tissue were isolated for five papers and ligaments MSCs (ACL) for two papers. Finally, all the studies used similar culture conditions, which allowed us to compare the data they extracted with each other. They isolated MSCs by processing different tissues, washing them, digesting them in collagenase, filtering them, washing the pellet, and then resuspending them in a fresh medium and plating the MSCs in flasks under tissue culture conditions. The non-adherent cells were removed after 48 h. The medium was changed on different days for each study passage when confluent enough (Table 1).

### 3.2. Proliferation of MSCs

Most studies assessed (11 out of 14) found an association between increased age of donor source of MSCs and diminished initial proliferative rate, as well as the capacity to maintain proliferation and clonogenic potential. Three studies did not identify proliferative or clonogenic differences between aged and young donor MSCs, [57,58,59] possibly due to the protocols for isolation and/or maintenance of MSCs having rate-limiting effects (Table 2).

**Table 1 ijms-24-15494-t001:** General characteristics of the studies.

References	Brief Description of Study	Source of MSCs	Number of Subjects	Age	Culture Conditions
Scharstuhl et al., 2007 [56]	BM-MSCs were isolated from the femoral shaft at total hip replacement.	BM - femoral shaft	98	24–92 years	In total, 25 mL of BM was collected, and the mononuclear cell fraction isolated through density gradient centrifugation. After 48 h, non-adherent cells were removed.
Stolzing et al., 2008 [60]	Investigated MSC from donors of various ages and determined their “fitness” by measuring various age and senescence markers in relation to their differentiation capacity and functionality.	BM - posterior iliac crest	57	Group I: 19–40 years old—“adult” group, Group II: >40 years old—“aged” group used in our analyses. Group III: 7–18 years old—“young” group not included in our review.	Lympho-prep was used for BM MNC separation and cryopreservation in liquid nitrogen. In the CFU-f test, 5 × 10^6^ BM MNCs were initially plated. Subsequent passages of MSCs were cultured at 1 × 10^6^ cells in T75 culture flasks.
Alm, J. J. et al., 2010 [61]	The study examined MSCs in fracture patients looking at cell surface markers, proliferation through several passages as well as osteogenic, chondrogenic and adipogenic differentiation.	BM - posterior iliac crest and Peripheral Blood (PB)	41	Group I: (76–95) Group II: (75–85) (19–60)	MNCs isolated, plated at 2 × 10^6^ (BM) or 5 × 10^6^ (PB) cells in 25-cm^2^ flasks. Non-adherent cells discarded after 48 h. Cells trypsinized after 14–21 days and re-plated at 1000 cells/cm^2^ in flasks.
Fickert et al., 2011 [62]	The study investigated the influence of donor age on proliferation and osteogenic differentiation in long-term ex vivo cultures of primary human MSCs from patients in different age groups.	BM - iliac crest	15	Group I: <50 yearsGroup II: 50–65 yearsGroup III: >65 years	Density gradient used and MNCs isolated.
Alt, E. U. et al., 2012 [63]	Adipose-tissue-derived MSCs (ASCs) were isolated from young, middle age, and aged healthy volunteers to investigate the effect of ageing on the self-renewal and differentiation potential of ASCs	ASCs - abdominal adipose tissue	40	15–71 Group I: <20 yearsGroup II: 30–40 yearsGroup III: >50 years	In total, 50 g tissue digested with collagenase I, RBC lysis buffer used. ASCs From three groups plated at densities of 1000 to 25 cells/cm^2^ in 12-well dishes. Analysis at day 10.
Siegel et al., 2013 [64]	BM-MSCs were assessed for phenotype, in vitro growth, colony formation, telomerase activity, differentiation capacity, T cell proliferation suppression, cytokine and trophic factor secretion, and receptor expression. Expression of Oct4, Nanog, Prdm14, and SOX2 mRNA was compared to pluripotent stem cells.	BM	53	13–80 years	Isolated mononuclear cells seeded at 1 × 10^5^ cells/cm^2^ in standard culture medium with 10% pooled human AB serum.
Ding, D.-C. et al., 2013 [57]	ASC isolated from abdominal subcutaneous fat of women undergoing gynaecological surgery.	Adipose Tissue Abdominal subcutaneous fat	27	Group I: (30–39 y) Group II: (40–49 y) Group III: (50–60 y)	ASCs dissociated with collagenase, passaged at 80% confluence, 1:3 ratio.
Choudhery, M. S. et al., 2014 [65]	Assessed effects of age on ASC expansion and differentiation. Measured expression of p16 and p21, population doublings (PD), superoxide dismutase (SOD) activity, cellular senescence, and differentiation potential.	Adipose Tissue	29	Group I: (<30) Group II: (35–55 y) Group III: (>60)	ASCs isolated via enzymatic digestion and then plated.
Ruzzini, L et al., 2014 [66]	Tendon stem cells (TSCs) were isolated through magnetic sorting from the hamstring tendons of six patients. TSC percentage, morphology and clonogenic potential were evaluated, as well as the expression of specific surface markers.	Hamstring tendons	6	Group I: (20–22) Group II: (28–31) Group III: (49–50)	Tendon biopsies yielded MSCs from fat and muscle via digestion and centrifugation, then cultured.
Lee, D.-H. et al., 2015 [67]	This study assessed the phenotypic and functional differences in ACL-MSCs isolated from younger and older donors and evaluated the correlation between ACL-MSC proportion and donor age.	ACL remnants from ACL reconstruction or TKA	69	Group I: Young—ACL reconstruction—29.67 ± 10.92 years Group II: Old—TKA—67.96 ± 5.22 years	Isolated ACL fascicles, washed, minced, digested, and plated after centrifugation and filtration.
Marędziak, M. et al., 2016 [68]	The study evaluated fibroblast colony forming unit (CFUF) count, proliferation rate, population doubling time (PDT), and lineage-specific differentiation parameters (osteogenic, adipogenic, chondrogenic).	Subcutaneous adipose tissue.	32	22 to 77 years old Group I: >20 years (mean age 24 ± 1.4 years; *n* = 8), Group II: >50 years (mean age 57.5 ± 0.7 years; *n* = 8), Group III: >60 years (mean age 67 ± 1.4 years; *n* = 8), and Group IV: >70 years (mean age 75 ± 2.8 years; *n* = 8).	Tissue samples digested in collagenase, centrifuged, and cells resuspended in culture medium for culturing.
Kawagishi-Hotta, M et al., 2017 [69]	ASCs were assessed for proliferation, as well as adipogenic, osteogenic, and chondrogenic differentiation potentials in vitro. Individual donor characteristics were analyzed via principal component analysis (PCA) based on these parameters.	Adipose tissue	260	5–97 years old	Subcutaneous adipose tissue digested to obtain SVF cells and ASCs, cultured in FGF-supplemented medium.
Prall, W. C. et al., 2018 [59]	This study found similar properties in hMSCs from iliac crest and proximal tibia, including proliferation and differentiation capabilities.	BM - MSCs from iliac crest or proximal tibia	46	Group I: young (18–49 years) Group II: Aged (≥50 years)	MSCs were isolated by washing the bone graft material and digesting it.
Andrzejewska, A. et al., 2019 [58]	Compared adult and elder BM-MSCs from a biobank, evaluating growth kinetics, gene expression, and differentiation potential.	Metaphyseal Bone Marrow - Core facility “Cell Harvesting” of the BIH Center for Regenerative Therapies (BCRT)	23	Mean age of adults = 38 years, Mean age of elderly = 72 years	Isolatedvia centrifugation and density grdients, cultured under standard conditions in an expansion medium.

### 3.3. Characterization of MSCs

All papers characterized the cells they processed. In the majority of the papers, characterization was performed using flow cytometry. Thirteen studies stained for CD90, a well-accepted MSC cell surface marker. One study did not stain for CD90 but stained for CD44, CD146, and STRO1 [66]. This study investigated the absence of CD34, as it proved a non-hematopoietic correlation (Table 2). Eleven papers used CD105, and eight papers used CD73. Based on the ISCT criteria, nine studies stained for negative markers CD34 or CD45. Some papers mentioned additional positive markers and some other negative markers. Some studies found no correlation between MSC phenotype and donor age, [56,57,58,59,62,63,65], whereas some studies indicated a correlation between MSC cell surface markers and donor age. In one study, young patients observed high levels of increased expression of MCAM, VCAM-1, ALCAM, PDGFRβ, PDL-1, Thy1, and CD71 [64]. A study that investigated CD44, CD90, CD105, and Stro-1, presented significant age-related changes such as increased expression levels of CD44 and a decrease in other cell surface markers [60]. Another study presented BM-MSCs and PB-MSCs from three younger fracture patients being less than 1% positive for CD45, CD14, CD19, and HLA-DR, and 98% positive for CD73, CD105, and CD90 [61]. A higher expression of CD73 was observed in the youngest group of patients in comparison to all the other groups, which indicated a variation connected with donor age and CD73. However, no differences were found with regard to the percentage of MSC surface antigen expression [68]. On the other hand, another study presented a higher expression of CD73 on ASCs from elderly donors compared to young donors. A significantly lower expression of CD105 was observed in ASCs of the high proliferation of young group donors compared to those in the low-potential group [69] (Table 2).

### 3.4. Other Outcomes

The ability of BM-MSCs to self-replicate was positively related to the expression of the Prdm14 mRNA (*n* = 18). The presence of Oct4 (*n* = 24) or Nanog mRNA (*n* = 24) was related to the lack of self-replicability of BM-MSCs [64]. In these cells, markers suggestive of senescence of the cells were noted with increased levels of ROS levels and increasing oxidative damage. These resulted in p21 and p53 levels increasing resulting in reduced MSC numbers and subsequently loss of differentiative potential. Increasing age resulted in progressively increased levels of NO, whilst superoxide dismutase (SOD) activity was reduced with increasing age. Biomarkers of ageing in MSCs included evidence of carbonylation of protein, and levels of lipofuscin and AGEs, which were markedly increased with increasing age and utilised to assess levels of oxidative stress of MSCs. Cellular homeostasis controlled by heat shock proteins (HSPs) was disrupted with increasing age as a result of reduced HSP levels. When passaged at the beginning, MSCs, irrespective of age, exhibited morphology that was similar in nature with spindles prominently present during replication. These are lost with further passaging as cells and the properties of the cells change with increased cell size [60]. Typically, samples from the cells in older tissue had less viable mononuclear cells [65] (Table 2).

### 3.5. Chondrogenic Differentiation of MSCs

In contrast to the findings in the majority of studies that suggest a clear link between aged donor MSCs and diminished clonogenic and proliferative potential, no such consensus exists with regard to the capacity for chondrogenic differentiation (Table 3). This is consistent with current thinking that while MSC donor age plays a role in chondrogenic differentiative capacity, other factors are also important. This explains why many of the studies that have been explored do not show homogeneity in their results (Table 3). Of the 14 studies identified, only 4 demonstrated reduced chondrogenicity with age, 5 showed no difference or no statistical difference, and 5 did not report any results after investigating the chondrogenesis.

### 3.6. Adipogenic and Osteogenic Differentiation of MSCs

Similar to the observation for chondrogenic differentiation, analyses of the impact of donor age on adipogenic (12 studies) and osteogenic (13 studies) differentiation of MSCs, did not yield a consensus (Table 4). Again, while it is likely that donor age and isolation/maintenance procedures play a role in differentiative capacity, other factors may be important. Finally, Scharstuhl et al. [56] did not investigate adipogenic or osteogenic differentiation. One study did not investigate the adipogenic differentiation [62]. From the 14 studies within our literature search, 4 papers demonstrated adipogenic differentiation reduction with age, 8 showed no difference or no statistical difference and 2 failed to report adipogenicity. With regards to osteogenic potential, six studies demonstrated a significant reduction in osteogenesis with age, seven studies did not demonstrate any meaningful difference in potential, and one failed to report osteogenecity.

### 3.7. Quality of Studies

A modified version of the OHAT tool was used to grade each study using some of the 11 questions listed in the Table below. Overall, eight studies were differently low risk. However, all 14 studies had probably a high risk of the “Blinding of research personnel” (Table 5). There was some concern towards the “Accounting for important confounding/modifying variables“ for four studies [67], and in five studies, there was concern towards other potential threats regarding “internal validity (bias)“. Overall, none of the studies were definitely high risk; all 14 studies included in this review were of low risk and high quality.

## 4. Discussion

### 4.1. Risk Factors of Osteoarthritis

Although the aetiology of osteoarthritis is not fully understood yet, which is a limitation for fully treating OA, there are risk factors that lead to osteoarthritis. The risk factors of OA could be categorized into modifiable, known as the secondary form of osteoarthritis, and non-modifiable, which represents the primary form of OA [70]. Unfortunately, the mechanism of the primary form of OA is not well understood. Heredity and age are both related to the primary form of osteoarthritis. Genetic predisposition plays an important role in the development of OA in its primary form [71]. On the other hand, secondary OA is mainly due to insults to the native joint such as articular injury, and predisposes the individual to developing secondary osteoarthritis through a culmination of variation in joint loading (biomechanical changes) as well as the biochemical milieu during acute injury which can be deleterious to cartilage [72].

Age is a well-accepted risk factor for OA. With the increase in the age of a patient, there is a higher risk of developing OA. Many studies consider weight and in particular, obesity as one of the highest risk factors for osteoarthritis as a result of the pressure and forces on the synovium and cartilage. Sex appears to play an important role in the risks of osteoarthritis. Females are at higher risk of OA, especially those who have undergone menopause. Although genetic risk factors are important, these have not been extensively explored [73]. Physical activity or intense exercise increases OA and can be compounded by intra-articular injuries (be they meniscal or ligamentous in nature). Although heavy physical workload did not demonstrate any significant proof for causing osteoarthritis, occupations that require kneeling and lifting appear to be a risk factor for OA [73]. Some occupations, like construction work and farming, have a high risk of causing future abnormalities to the knees of a patient [71]. Lifestyle could be considered a risk factor as well. For example, smoking varies as a risk factor with equivocal results with some studies presenting it as a protective factor and some as a risk factor for future OA [71].

### 4.2. Chronological Age and MSCs Behavior In Vitro and In Vivo

In vivo, ageing varies in comparison to in vitro ageing in its definition. In vivo, ageing is represented by the chronological age of a donor, whereas in vitro, this is typically represented by the loss of stem cell characteristics as cell lineage differentiates into a particular phenotype during expansion. The earlier the passage of the cells, the closer the environment is to the human body it is, and therefore, cells behave under similar conditions. Colony formation ability, osteogenic potential, and proliferative capacity are reduced in older MSC donors in comparison to younger ones. Most studies suggest a direct correlation between prefiltration and MSC donor age. The proliferation potential decreases with the increase in age. The rate of proliferation in vivo is low and there is no de-differentiation of chondrocyte reported. In vitro, they are cultured monolayer, and they are lost in the twenty-day phenotype [74]. Some investigations lead to a correlation between senescence and donor age of MSCs; one of the reasons might be the elevated expression of the *BAX* gene and p21 and P53 (its pathway genes) [75]. Cartilage, like all other tissues, presents changes with the increase in chronological age. Decreases in mitosis and metabolic activity are observed due to the extinction of the blood vessels. Aged cartilage is thinner because of declining proteoglycan levels and increased collagen crosslinking [76].

### 4.3. Effects—MSCs Characterization

The studies we reviewed here characterized the cells as MSCs using some of the ISCT positive and negative cell surface markers. However, a few studies did not include all the minimum criteria that ISCT proposed. An MSC population that included some, but not all the minimal criteria, was described and presented by several studies included in this review. Finding an optimum, unified definition of what characterizes MSCs through identifying specific surface markers and functional assays will provide homogeneity between studies and would be favourable for future studies. The heterogeneity seen in these studies means that there are varying opinions of what defines an MSC and can add extra variables to these studies.

### 4.4. Effects—Chondrogenic Potential

Chondrogenic potential is compromised with age, although this finding is not uniform between studies. Only 4 of 14 studies state that with an increase in age, there is a decrease in the chondrogenic potential of MSCs [60,63,65,68]. What causes this loss of potential is speculated to involve the difficulty in stimulating genes, allowing for chondrogenic growth factors and an environment conducive to these cells [77]. The interplay with other factors may represent why there is a lack of equipoise with regard to the influence of age in chondrogenesis. Then, different factors may influence the chondrocyte differentiation in addition to just age. Identifying these factors remains an area of current research. The role of immune cells within these regions is believed to influence the ability of an MSC to show chondrogenic potential and may represent the damaged tissue within cartilage. Identifying these particular biomarkers will become important in helping us better understand which biochemical pathways influence MSCs’ aid with cartilage repair and will provide a useful tool in patient stratification in the future. This may also explain the heterogeneity in the results found. Further analysis in understanding the differentiating genes between different aged donors can help reviewers understand the differential gene profile through transcriptomics and metabolomics of MSCs.

### 4.5. Adipogenic and Osteogenic Potential

Four studies we reviewed reported a correlation between adipogenic potential and age; six reported a correlation between osteogenic potential and age. The potential of osteogenic and adipogenic differentiation is decreasing with the increase in the age of MSC donors. This may point to similar mechanisms influencing both of these and may represent a biochemical pathway that works in tandem with each other, depending on the state of the neighbouring cells. Some of the studies identified within our study appear to conclude that this regenerative potential is diminished with age. This is likely a result of cumulative mutations within these cells that drive the cell into an apoptotic pathway and resultant cell death [78]. It is important to note that not all studies observed this relationship. The majority of the reviewed papers did not find any significant effect of age on the differentiation capabilities of MSCs into adipocytes and osteoblasts. This divergence in findings could be attributed to variations in study design, sample size, cell culture conditions, or other factors that may influence the observed outcomes. Reversing these to prevent this from activating and potentially upregulating these mechanisms will be further identified with the use of novel genomics, metabolomics, and transcriptomic technology and provide a potential avenue for researchers to explore in the future.

## 5. Conclusions

This systematic review demonstrates a heterogeneous conflicting range of outcomes concerning the chondrogenic, osteogenic, and adipogenic potential of MSCs in relation to age. While several studies present a correlation when comparing chronological age with the osteogenic and adipogenic potential of MSCs within the literature, others present varied findings on this matter. In addition, most studies that we reviewed suggest a clear link between aged donor MSCs and diminished clonogenic and proliferative potential. This does not appear to be reflected in chondrogenesis in these cells, as only four reported a decline in the chondrogenic potential with increased age. Chronological age-related changes in MSC function have important implications for the use of these cells in clinical applications for an ageing population. Further investigations on the in vitro effects of chronological age on the chondrogenic potential of MSC should follow the outcomes of this systematic review, shedding more light on this complex relationship. The results from this study should be used to plan further investigations looking at the effects of chronological age on cellular senescence and identify pathways that could be targeted to potentially reverse any age-related changes. Understanding the pathomechanisms involved in the process can highlight potential reversal targets in promoting and maintaining cartilage repair.

Osteoarthritis continues to be an ever-increasing, debilitating ailment for patients and health professionals must be prepared to treat and provide them with differing ways for symptomatic and functional relief. A wave of strategies including MSC therapy could provide an avenue for the prevention of the development of primary and secondary OA and using new-age technology such as genomics, transcriptomics, and functional assays can help scientists target this disease once and for all.

## Figures and Tables

**Figure 1 ijms-24-15494-f001:**
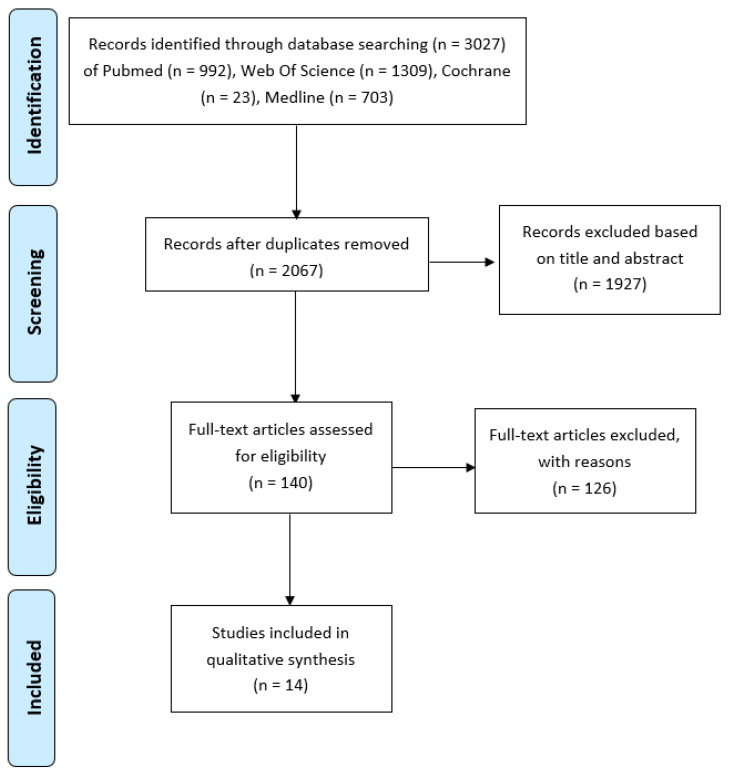
PRISMA Flow Diagram (Figure 1 in [54]).

**Table 2 ijms-24-15494-t002:** Proliferation and MSC characterization.

References	Proliferation Analysis	Proliferation Results	MSC Characterization	Other Outcomes
Scharstuhl et al., 2007 [56]	MSC count was estimated by counting adherent cell clones in a 10-cm^2^ dish 4 days after seeding 2 × 10^6^ mononuclear cells. Confluence was reached in 9–11 days, followed by 5 passages, resulting in approximately 25 population doublings per initial MSC.	Age showed no correlation with BM mononuclear cell count, MSC yield, or cell size. Proliferative capacity and cellular spectrum remained independent of age.	Primary MSCs from different groups consistently expressed CD10, CD73, CD90, CD105, CD109, CD140b, CD164, CD166 as confirmed by flow cytometry. No correlation was observed between MSC phenotype and donor age.	N/A
Stolzing et al., 2008 [60]	Age-related reductions were observed in CFU-fibroblast (CFU-f) numbers and in the counts of CD45^low/D7fib^+ve/LNGF^+ve cells. Additionally, there was diminished capacity for proliferation and differentiation.	Significant decline in CFU-f numbers was observed in the older age group. In terms of proliferation potential, all cultures showed similar initial growth, but after about 5 weeks, age-related differences emerged. Proliferation in the “aged” MSC cultures began to decline, reaching a plateau, while cultures from “adult” donors continued to proliferate.	Passages 1–5 MSC were stained for various markers. They consistently expressed CD13, CD44, CD90, CD105, Stro-1, and D7-Fib regardless of age. Significant age-related changes were observed in the expression levels of CD44 (increase) and CD90, CD105, and Stro-1 (decrease).	Indices of cellular ageing including oxidative damage, ROS levels, and p21 and p53 all increased suggesting a progressive loss of MSC numbers and differentiation capacity with age. NO increased progressively with age, and SOD activity declined significantly with age. Levels of oxidised proteins (carbonyls), AGEs and lipofuscin content (biomarkers of ageing) significantly increased with age. Age-related decrease in mean HSP levels with age.
Alm, J. J. et al., 2010 [61]	Population doublings (PDs) were calculated at each passage. Colony formation efficiency was assessed by plating cells at 100 cells/well in a 6-well plate and counting colonies after 21 days. Cell proliferation was monitored by calculating PD and PD rate at each passage (P).	Total PDs and PD rate were significantly higher for younger fracture patients (group III) compared to those of elderly patients (groups I and II). Higher number of colonies formed by MSCs from younger patients using CFU assay. Linear regression confirmed an age-dependent decline in total PDs.	Cells from groups I and III displayed positive expression for CD73, CD105, and CD90, while being negative for CD45 and CD14. In a limited flow cytometric analysis, cells from three younger fracture patients (group III) showed 98% positivity for CD73, CD105, and CD90, with less than 1% positive for CD45, CD14, CD19, and HLA-DR.	N/A
Fickert et al., 2011 [62]	After the expanded cells reached ~80% confluence in P0, adherent cells from split in P1 into four similar plates. MSCs were seeded at low density (2 × 10^5^) in a new culture flask for future expansion. From P1–8, every expansion flask was split to four similar plates. Stop of proliferation was defined by more than the double mean time period of former cultivation time and morphological changes such as polygranulation and polynucleation.	MSC proliferation time varied by patient group. Cells from young and elderly groups (I and III) grew faster (5 and 7 months) than those from the middle-aged group (II; 11 months), although cells in groups II and III showed a wide range of individual doubling times.	Regardless of age, over 60% of MSCs expressed CD166, CD105, CD90, CD54, and CD73 after P1. CD166 was present in 90–100% of cells regardless of differentiation stage or age group. No significant differences were observed in CD marker expression between expanded and differentiated MSCs.	N/A
Alt, E. U. et al., 2012 [63]	For doubling time experiments, 20,000 ASCs from each group were plated in a 75 cm^2^ flask and counted at 48, 72, and 96 h. The population doubling time was calculated from at least three time points, and the mean was determined.	A weaker CFU ability and increased population doubling time were observed in groups 2 and 3 compared to group 1.	Flow cytometry analysis showed consistent expression of CD44, CD90, CD105, and CD146, and absence of CD3, CD4, CD11b, CD34, and CD45 surface markers in ASCs from all three groups.	N/A
Siegel et al., 2013 [64]	Colony formation capacity was assessed by seeding sub-confluent primary BM-MSCs (P0) at densities of 100, 200, and 500 cells per well in six-well plates at P1. After 10 days of culture, cells were fixed and stained. The percentage of colony formation was calculated for each seeding density and MSC preparation.	At P1, a density of 1000 cells per cm^2^ was used. Subpopulations of more rapidly dividing cells expressed surface markers at a higher density. While no correlation was found between donor age and MSC proliferation capacity (*n =* 52), high clonogenic BM-MSCs, which were smaller, divided more rapidly, and were more frequent, were observed in preparations from younger female donors.	BM-MSCs from younger donors exhibited elevated expression of MCAM, VCAM-1, ALCAM, PDG-FRβ, PDL-1, Thy1, and CD71. Conversely, CD71, CD90, CD106, CD140b, CD146, CD166, and CD274 showed a negative correlation with donor age.	The clonogenic potential of BM-MSCs did not align with lineage-specific mRNA expression. Clonogenic potential correlated positively with Prdm14 mRNA expression but not with Oct4 or Nanog mRNA. The levels of Oct4, Nanog, and Prdm14 mRNA in BM-MSCs were notably lower compared to pluripotent stem cells and were independent of donor age.
Ding, D.-C. et al., 2013 [57]	ASCs at P2–3 were used for proliferation assay. Cells were harvested and counted on days 0, 2, 3 and 4, and a growth curve generated. To calculate the PD time, 1 × 10^4^ cells were seeded in a Petri dish.	Proliferation capacity was not reduced in the older population groups. The average PD time for all ASC donors was 21.5 ± 2.3 h.	Surface expressions of CD13, CD44, CD90 and HLA-ABC consistent with BMSC. CD34 was expressed in 11.9 ± 8.8%, 9.8 ± 7.7% and 16 ± 4% of ASC derived from 30–39 y, 40–49 y and 50–60 y age groups, respectively.	ASCs exhibited a similar immunophenotype to BM-MSCs. Chronological age was linked to increased pre-adipogenic commitment and reduced adipogenic differentiation potential, while other characteristics remained consistent.
Choudhery, M. S. et al., 2014 [65]	Cumulative growth index. Serial passaging at 1:10 dilution, measuring cells before and after each passage. The population doublings (PDs) and doubling time (DT) were calculated.	Higher population doublings in older age groups: PD = 44.1 ± 7.1 vs. 38.5 ± 4.3 vs. 34.3 ± 8.1 doublings for young, adult, and aged donors, respectively. Additionally, longer doubling time observed at older ages: 62.0 ± 5.9, 80.9 ± 29.6, and 89.1 ± 26.6 h for young, adult, and aged donors, respectively.	Flow cytometry; MSCs were strongly positive for MSC markers (CD44, CD73, CD90, CD105) while lacking expression of hematopoietic markers (CD3, CD14, CD19, CD34, CD45).	Aged ASCs showed reduced viability, proliferation, and increased senescence. Additionally, their differentiation potential was reduced. Tissue samples from older donors also yielded fewer viable mononuclear cells.
Ruzzini, L et al., 2014 [66]	Colony Forming Assay: TSCs were seeded at 2 cells/cm^2^ and cultured for 15 days. Colonies >2 mm in diameter were counted and sized after staining with 1% crystal violet.	The size of the colonies formed by group 3 (older) was significantly larger than the colonies formed by the other groups.	Tendon-derived CD44+ cells expressed stem cell markers CD146 and STRO1. They were negative for CD34, confirming absence of hematopoietic cell contamination. These cells also exhibited tenogenic markers a-SMA and TNMD.	Effects of age on proliferation but not differentiation potential.
Lee, D.-H. et al., 2015 [67]	1 × 10^5^ primary ACL-derived MSCs were cultured for 14 days, detached, and counted. Subsequently, 1 × 10^5^ P1 cells were seeded into a 100-mm diameter culture dish and harvested after 7 days to obtain P2 cells.	At passage 0, the mean proportion of MSCs was significantly higher in ACL cells from the TKA than from the ACL reconstruction group (19.69 ± 8.57% vs. 15.33 ± 7.49%). However, MSC proportions at P1 and P2 were similar in the two groups.	MSCs were defined as cells triple positive for CD44, CD90 and CD105 and negative for CD34. Acquired cells were gated as P1 and surface expression of CD44, CD90, and CD105 was assessed in CD34–population.	Gene profiles of P2 MSCs from both groups were analyzed using microarray analysis, revealing 40 genes with 2- to 18-fold differential regulation. In the older group, the top three genes with higher expression were *C7orf28B*, *XIST*, and *PRG4*. Conversely, the top three genes with reduced expression in the older group were *RPS4Y1, PSG5*, and *EIF1AY*.
Marędziak, M. et al., 2016 [68]	The proliferation of hASC was assessed using a commercial kit on the 2nd, 5th, and 7th days. Proliferation factor and population doubling time (PDT) were calculated from absorbance measurements.	Group 1 cells exhibited higher proliferation rates, reaching full confluence by day 5. Older donor samples (groups 2, 3, and 4) showed similar growth curves. PDT correlated with age, with younger donors achieving it faster. CFU formation ranged from 0.9% (>70) to 5.7% (>20) and was age dependent.	Flow cytometry analyzed MSC surface markers (CD44, CD73, CD90, CD105, CD34, CD45). While overall expression percentages remained consistent, CD73 showed slight age-related variation, with higher expression in the youngest group compared to others.	Aged hASCs had increased senescent features.
Kawagishi-Hotta, M et al., 2017 [69]	Cultured up to passage 5, PDL and doubling time were assessed by viable cell counts using a hemocytometer. Seeding density maintained at 1–2 × 10^4^/cm^2^, with media renewal every 2–3 days.	The doubling time of p4 ASCs increased with age, but no significant correlation was observed between proliferation potential and donor age (*r* = 0.099).	Elderly group cells showed higher CD73 expression compared to the young group. In the young group, highly proliferative cells had lower CD105 expression than low-potential cells.	N/A
Prall, W. C. et al., 2018 [59]	Passage 1 cumulative PD and PDT were evaluated over 8 weeks. Clonogenic efficiency was determined using the CFU assay.	Cumulative PD for iliac crest: Young—17.6 ± 1.9, Old—14.8 ± 5.2. Cumulative PD for proximal tibia: Young—17.0 ± 2.8, Old—17.4 ± 3.1. No significant PDT differences between sites or age groups. Proliferation peaks between day seven and ten, then decreases by day 14.	The cells derived from all donor sites showed a positive expression of CD73, CD90 and CD105.	N/A
Andrzejewska, A. et al., 2019 [58]	BMSC growth kinetics were quantified by calculating population doublings at each passage.	BMSCs showed consistent growth kinetics at passages 3 to 6, irrespective of donor age or diabetic status. Increased cell diameter and volume were observed only in the elderly at P6.	No age-based differences were observed. The isolated cells expressed typical MSC markers (CD73, CD90, CD105, and CD146) while being negative for contaminating cell populations (CD14, CD19, CD31, CD34, and CD45)	No significant difference found.

PD = population doubling; PDT = population doubling time; P = passage; CFU = colony forming unit assay; N/A= Not Applicable.

**Table 3 ijms-24-15494-t003:** Chondrogenic differentiation.

References	Protocol	Gene Expression	Histological Staining	Other Biochemical Assays	Result
Scharstuhl et al., 2007 [56]	A droplet containing 4 × 10^5^ primary cells in CM supplemented with TGF-β3 cultured for 21 days.	*- ACAN* (aggrecan) *- COL1* and *COL2* (collagen) No correlation between mRNA profiles and donor age could be detected.	Safranin-O/Fast Green staining, type II and type I collagen immunohistochemistry. Hypertrophy determined by staining did not correlate with age.	N/A	MSCs differentiated into the chondrogenic lineage Irrespective of age.
Stolzing et al., 2008 [60]	2.5 × 10^5^ cells cultured in CM for 21 days.	N/A	Sulfated GAG was visualized on Toluidine Blue and Alcian Blue binding assay.	N/A	Chondrogenic differentiation declined in “aged” MSC compared to “adult” MSC although this did not reach significance.
Alm, J. J. et al., 2010 [61]	MSCs (2 × 10^5^) were pelleted and cultured in CM	N/A	After 3 weeks, Toluidine Blue staining showed a proteoglycan-rich extracellular matrix and the presence of chondrocyte-like lacunae in all groups.	N/A	N/A
Fickert et al., 2011 [62]	Expanded MSCs (1 × 10^6^) were cultured in pellet form with TGF-β3 for 3 weeks for subsequent gene expression, histological, and immunohistological analyses.	RT-PCR gene expression analysis using *COL2* and *COMP*	Similar Alcian Blue staining in a 55-year-old patient and a 73-year-old patient.	N/A	N/A
Alt, E. U. et al., 2012 [63]	Chondrogenic differentiation with about 1 × 10^5^ cells spun in a 15 mL conical tube and grown in CM for 21 days.	Down-regulation of *BMP6, COL2A,* and *Col10A* genes observed in cells from group III compared to group I.	Micro masses were fixed and stained with Toluidine Blue	N/A	Chondrogenic potential decreased with age.
Siegel et al., 2013 [64]	Differentiation induced with hMSC Chondrogenic Differentiation BulletKit (PT-3003, Lonza) + TGF-β3.	No significant differences detected in lineage-specific mRNA expression for chondrogenesis (*SOX9*, *n =* 47 and *COLL2, n =* 32).	Frozen sections of fixed pellets were stained with Safranin O for chondrogenesis (*n =* 40)	N/A	No donor age related differences detected.
Ding, D.-C. et al., 2013 [57]	ASC at P3 isolated and grown in chndrogenic media containing TGF-β1 at density 1 × 10^5^. After 3w, fixed in slides and stained using standard Alcian Blue protocols.	qRT-PCR for gene expression at Day 0 and Day 21 of *ACAN and COL2*	Alcian Blue	Size of pellet/micromass	No effect of chronological age on chondrogenic potential.
Choudhery, M. S. et al., 2014 [65]	Chondrogenesis was induced in micromass pellets derived from 2.5 × 10^5^ MSCs, cultured in chondrogenic medium for 3 weeks.	qRT-PCR of: -*ACAN* (aggrecan) -*COL2* (collage type-2)	Alcian Blue	N/A	ASC chondrogenic potential declines with donor age. Stronger Alician Blue staining and higher levels of expression of lineage-specific markers.
Ruzzini, L et al., 2014 [66]	Seeded at 1 × 10^5^ density. CM containing TGF-b2 for 3 weeks	*SOX-9* was expressed significantly more in group III compared to the other two groups and was expressed significantly more in group I compared to group II.	Alcian Blue staining showed strong positivity for acid mucopolysaccharides and epithelial mucins/cartilage.	N/A	N/A
Lee, D.-H. et al., 2015 [67]	Group III (49–50 years) showed significantly higher SOX-9 expression compared to groups I (20–22 years) and II (28–31 years).	N/A	Toluidine Blue: there were no significant differences in absorbance at 595 nm between the 2 groups (0.32 ± 0.13 versus 0.45 ± 0.11, *p* = 0.258).	Comparison of chondrogenic pellet sizes showed that there were also no significant differences between the 2 groups (1.60 ± 0.65 mm versus 2.10 ± 0.37 mm, *p* = 0.121).	N/A
Marędziak, M. et al., 2016 [68]	Chondrogenic Differentiation in 24-well plates and inoculated at concentration of 30 × 10^3^ cells per well. The media was changed every two days for 21 days.	Col II expression was higher in the youngest group compared to the older groups, while Aggrecan expression was higher in the youngest group compared to the oldest.	Safranin O	Collagen II levels were higher in the youngest group compared to all other groups, measured by ELISA.	The ability of hASCs to differentiate into chondroblasts decreases with age.
Kawagishi-Hotta, M et al., 2017 [69]	Passage 5 cells (1 × 10^5^) were centrifuged and suspended in CM with BMP-2 and TGF-b1.	N/A	Normalized GAG did not show a correlation with age (r = 0.059), but a large variation was noted.	CFU-F assay. Chondrogenesis was assayed by the sulfated GAG content and normalized with DNA content.	Chondrogenic potential of cells was not correlated with donor age, but individual differences were observed in all age groups.
Prall, W. C. et al., 2018 [59]	Cells were preconditioned in hypoxia for four days during monolayer expansion. Pellets containing 4.5 × 10^5^ cells were stimulated for 28 days in conditioned medium supplemented with TGFβ1 and BMP2.	N/A	Safranin O staining: Age-pooled chondrogenic differentiation showed 82.8% ± 27.0 for iliac crest-derived cells and 88.3% ± 22.9 for proximal tibia-derived cells.	N/A	No significant different differences were observed between both age groups.
Andrzejewska, A. et al., 2019 [58]	Passage 6 cells were cultured in V-bottom 96-well plates with TGF-β3 enriched culture medium for up to 21 days.	N/A	Alcian Blue staining of pellet sections showed similar proteoglycan production in BMSCs from adult and elderly donors upon chondrogenic differentiation (*p* < 0.01 and *p* < 0.001).	N/A	N/A

CM = chondrogenic media; qRT-PCR = quantitative reverse transcription polymerase chain reaction; GAG = glycosaminoglycan.

**Table 4 ijms-24-15494-t004:** Osteogenic and adipogenic differentiation.

Reference	Protocol	Results	Protocol	Results
Scharstuhl et al., 2007 [56]	N/A	N/A	N/A	N/A
Stolzing et al., 2008 [60]	Cells were incubated in OM for 10 days. ALP quantification was performed using a colourimetric assay. Changes in VDR, GR, and the Notch-1 receptor.	ALP activity declined with age, with “aged” MSCs displaying significantly lower activity compared to “adult” MSCs. Notch-1 and VDR levels also showed a significant decrease with age, while GR levels increased significantly in “aged” MSCs compared to ‘‘adult’’ MSCs.	Adipogenic differentiation was assessed by quantifying the percentage of Oil Red O-positive cells after culturing in adipogenic medium (AM).	Age did not lead to significant changes in the percentage of Oil Red O-positive cells.
Alm, J. J. et al., 2010 [61]	ALP and von Kossa staining at 2 and 4 weeks, respectively, along with calcium content measurement, were conducted to assess osteogenic differentiation.	Elderly-patient-derived cells exhibited reduced osteogenic potential compared to younger patient-derived cells. This was evident in lower ALP expression and mineralization. The decline in ALP expression and von Kossa staining was age-dependent.	Cells were cultured in AM for 3 weeks and evaluated for accumulation of intracellular lipid droplets.	Both BM-MSCs and PB-MSCs displayed increased lipid droplets after 21 days of differentiation, indicating adipogenic differentiation post-passage. However, quantitative analysis was not conducted.
Fickert et al., 2011 [62]	Cells were cultured in osteogenic medium for 11 days, and osteogenic differentiation was assessed through ALP activity and gene expression analysis of osteogenic markers (Coll I and II, Cbfa1, ALP, OC, and BSP1).	With increasing donor age, there is no observed reduction in ALP activity. In passage 1, ALP activity was approximately 465 mU/mg in group I, 283 mU/mg in group II, and 344 mU/mg in group III. By passage 2, ALP activity decreased in the youngest donor group and increased in the older groups. Overall, the highest ALP activity was detected in the age group over 65 years. The detectable frequency of genes differed; however, it was independent of donor age. Osteogenic markers in all groups increased over several passages, with group III exhibiting the highest expression level in passage 1.	N/A	N/A
Alt, E. U. et al., 2012 [63]	Differentiated cells were stained with Alizarin Red or quantified for ALP.	Osteogenic differentiation potential declined significantly with age. ALP concentration in group 3 was markedly reduced when compared to group I (from ~50% in group I to ~22% in group III).	Adipogenic differentiation evaluated by Oil Red O staining and real-time PCR analyses of lineage-specific genes.	The percentage of cells undergoing adipogenic differentiation decreased from approximately 33% in group I to about 10% in group III. Real-time PCR analysis of LPL and CEBPA on lineage-specific transcriptomes showed a down-regulation in group III compared to group I.
Siegel et al., 2013 [64]	OM with dexamethasone, ascorbic acid and β-glycerolphosphate. Osteogenesis assessed by Alizarin Red staining (*n =* 40) and lineage-specific mRNA expression of OPN (*n =* 17) and AP (*n =* 41).	No donor age-related differences detected in the osteogenic differentiation for Alizarin Red staining (*n =* 40). No statistically significant differences for the lineage-specific mRNA expression of OPN and AP.	Commercial adipogenic kit. Adipogenesis assessed by lineage-specific staining with Oil Red O staining (*n =* 40) and lineage-specific mRNA expression of LPL (*n =* 44) and PPARγ (*n =* 48).	No donor age-related differences were detected in the adipogenic differentiation capacity, analyzed by Oil Red O staining. Similarly, no statistically significant differences were observed in the lineage-specific mRNA expression of LPL and PPARγ.
Ding, D.-C. et al., 2013 [57]	ASC harvested at P3, cultured in OM for 21 days and stained with Alizarin Red. Gene expression assessed for osteopontin.	Osteogenic potential using gene expression for osteopontin, and Alizarin Red staining was not related to donor’s age.	P3 cells differentiated using AM. Adipogenesis and lipid vacuole formation in the ASC were studied by staining cells with Oil Red. Gene expression of PPAR-γ also assessed.	Older age (II, III) groups exhibited reduced adipogenic potential, with lower intracellular lipid content and decreased expression of the PPAR-γ gene compared to the I group.
Choudhery, M. S. et al., 2014 [65]	Osteogenic differentiation was induced using OM for 3 weeks. Assessment was conducted via Von Kossa staining and gene expression analysis of ALP and osteocalcin.	Osteogenic potential decreases with age. Young donors showed significantly higher expression of ALP and osteocalcin compared to adult and aged donors.	Adipogenic induction medium, for 3 weeks and assessed by Oil Red O staining.	Chronological age had no effect.
Ruzzini, L et al., 2014 [66]	OM for 3 weeks. Then fixed and stained with Von Kossa stain. Gene expression using RUNX-2	RUNX-2 expression was significantly higher in group I compared to the other two groups. Von Kossa staining showed clustered areas of calcium deposition.	Cells were cultured in adipogenic induction medium for 3 weeks, followed by fixation and Oil Red O staining. Additionally, PPARG mRNA expression was assessed.	PPARG mRNA expression was significantly higher in group I compared to the other two groups. Additionally, lipid droplets were observed in cells on Oil Red O staining.
Lee, D.-H. et al., 2015 [67]	Passage 1 or 2 cells were plated in a 24-well plate. Invitrogen OM used once cells were 50–70% confluent for 2 weeks with medium changes twice per week. Osteogenic analysis using Alizarin Red staining.	No significant differences in staining between groups (*p =* 0.547).	Passage 1 or 2 cells were cultured in MSCGM until 100% confluence, then switched to STEM-PRO AM for 2 weeks with bi-weekly medium changes. Adipogenic differentiation was assessed using Oil Red O staining.	No significant differences in absorbance at 490 nm between groups (*p =* 0.875).
Marędziak, M. et al., 2016 [68]	Osteogenic Differentiation Kit used in 24-well plates for 21 days with OM changed every two days. Osteogenic differentiation assessed using Alizarin Red staining, and BMP-2, ACAN, and Col-I ELISA, and Col-II, ADIQ, and LEP assays. Expression of osteoblast-specific markers (OPN, Col-I, OCL, and BMP-2) were analyzed by qRTPCR.	Osteogenic differentiation potential decreases with donor age. Gene expression of OPN, OCL and BMP-2 revealed higher expression in younger patients. For Col-I, a similar trend was observed across all donor groups.	Adipogenic differentiation was induced using the Adipogenic Differentiation Kit in 24-well plates with Adipogenic Medium (AM) for 14 days. Differentiation was evaluated through Oil Red O staining, and the concentration of adiponectin (ADIQ) and leptin (LEP) in the medium. Expression of adipocyte-specific markers (LEP, ADIQ, and PPAR-γ) was assessed by qRT-PCR.	No differences between age groups in Oil Red O staining. Leptin, adiponectin, and PPAR-γ concentrations were elevated in older patients as compared to the >20-year-old group.
Kawagishi-Hotta, M et al., 2017 [69]	Passage 5 cells were seeded in a 24-well plate After 4 days, differentiation with OM for 21 days with medium changed every 2–3 days. Osteogenic differentiation assessed using normalized quantification of Ca deposition and Alizarin Red staining.	Normalized calcium deposition exhibited no correlation with age (*r =* 0.005). While large variations were observed, individual differences were evident across all age groups. These differences in osteogenic potential increased incrementally with donor age.	Passage 5 cells were seeded in a 24-well plate. After 4 days, differentiation with AM for 4–6 days. Adipogenic analyses using Oil Red O.	Normalized concentration of eluted Oil Red O showed a significant correlation with age(*r =* −0.283). Individual difference in adipogenic potential was great and observed in all age groups.
Prall, W. C. et al., 2018 [59]	Passage 3 cells were fully confluent before differentiation. Alizarin Red staining was performed at 7 and 14 days. Gene expression of DLX5, RUNX2, ALPL, and SPP1 was assessed.	No significant differences. Comparable increase in gene expression in all donors. Alizarin Red staining after 14 days showed no difference between the groups.	Cells were seeded in 6-well plates and exposed to BODIPY AM for 5 days followed by 2 days in preservation media. This process was repeated for 21 days.	Adipogenic differentiation showed no significant differences between groups or donor sites.
Andrzejewska, A. et al., 2019 [58]	Osteogenesis was assessed using Alizarin Red	No difference between adult vs. elderly donors at day 14 or day 22 time points, and passage 3 and 6. A strong reduction in mineralization for higher vs. lower passage cells. Cells from adult and elderly donors exhibited significantly diminished osteogenic differentiation at passage 6 compared with passage 3 (*p* < 0.01 and *p* < 0.001).	Adipogenesis was assessed using Nile Red.	No difference between adult vs. elderly at either time point but a stronger passage-dependent reduction for cells from elderly donors compared to adult donors. Adipogenic differentiation potential only showed minor changes, mainly reduced lipid formation, when comparing early and late passages or at later readout (*p* < 0.05).

OM = osteogenic media; AM = adipogenic media; ALP = alkaline phosphatase; LEP = leptin.

**Table 5 ijms-24-15494-t005:** Risk of bias analysis/assessment.

	Clinical trial ID													
	1	2	3	4	5	6	7	8	9	10	11	12	13	14
Randomisation of administered dose or exposure level	N/A	N/A	N/A	N/A	N/A	N/A	N/A	N/A	N/A	N/A	N/A	N/A	N/A	N/A
Allocation concealment	N/A	N/A	N/A	N/A	N/A	N/A	N/A	N/A	N/A	N/A	N/A	N/A	N/A	N/A
Appropriate participant selection for comparison	++	++	++	++	++	++	+	++	-	+	++	++	++	++
Accounting for important confounding/modifying variables	+	-	-	-	+	+	++	+	+	-	+	+	+	+
Identical experimental conditions across study groups	++	++	++	++	++	++	++	++	++	++	++	++	++	++
Blinding of research personnel	-	-	-	-	-	-	-	-	-	-	-	-	-	-
Complete outcome data	++	++	++	++	++	++	++	++	+	++	++	++	++	++
Confidence in exposure characterisation	++	++	++	++	++	++	++	++	++	++	++	++	++	++
Confidence in outcome assessment (incl. assessor blinding)	+	+	+	+	+	+	+	++	+	+	+	+	+	+
Complete reporting of measured outcomes	++	++	++	++	++	++	++	++	++	++	++	++	++	++
Other potential threats to internal validity (bias)	+	-	-	++	-	++	+	+	-	-	+	+	+	+

1 = Siegel (2013) [64]; 2 = Scharstuhl (2007) [56]; 3 = Stolzing (2008) [60]; 4 = Alm (2010) [61]; 5 = Fickert (2011) [62]; 6 = Alt (2012) [63]; 7 = Ding (2013) [57]; 8 = Choudhery (2014) [65]; 9 = Ruzzini (2014) [66]; 10 = Lee (2015) [67]; 11 = Maredziak (2016) [68]; 12 = Kawagishi-Hota (2017) [69]; 13 = Prall (2018) [59]; 14 = Andrzejewksa (2019) [58]. ++ Definitely low risk, + Probably low risk, - Probably high risk.

## Data Availability

Not applicable.

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
