# Peer review of "The Effects of Chronological Age on the Chondrogenic Potential of Mesenchymal Stromal Cells: A Systematic Review"

_ijms, 2023, doi:10.3390/ijms242015494_

Round 1

Reviewer 1 Report

Dear Authors,

I read your review on "The effects of chronological age on the chondrogenic potential of mesenchymal stromal cells (MSCs): a systematic review" and found it interesting, however I believe it is necessary that the text is improved with appropriate changes according to the reviews that I show you below:

  • In the introduction, line 41-42, you describe the isolation of MSCs from other tissues, but dental pulp is completely missing among the various tissues of origin. You must integrate with a part dedicated to MSCs isolated from dental tissues (DPSCs, SHEDs etc..) and related bibliography (DOI: 10.1073/pnas.240309797; doi: 10.3791/59282; doi: 10.3390/biomedicines10123112).

·        Line 45 add more and recent references on neuronal differentiation.

  • In the introduction you talk about osteo, chondro and adipo differentiation, the neuronal differentiation is missing since you previously said that MSCs can differentiate into glial cells. Add a description.
  • I recommend you read the present article: Age of the donor affects the nature of in vitro cultured human dental pulp stem cells" (https://doi.org/10.1016/j.sdentj.2020.09.003), to improve the part where you talk about the effect of age on the functions of MSCs. Explain whether the effects of age are related to the age of the donor from which the MSCs are taken, or whether the effect depends on the time of in vitro culture at different passages, after isolation?

·        Why did you exclude periodontal and dental pulp MSCs in this study? Why did you exclude neurological differentiation potential? Can you justify your choice?

·        Format table 1 according to the journal style and standardize the presentation of the data collected. Make the table more streamlined and easier for readers to understand.

·        Tables 2-3-4 are too complex; I ask you to streamline them to facilitate understanding and reading.

·        Review the text, trying to reformulate the results part in a more concise manner.

  • Review the referencing style according to the Instructions for Authors and in the text, reference numbers should be placed in square brackets [ ] and placed before punctuation; for example [1], [1–3] or [1,3]. The reference list should include the full title, as recommended by the ACS style guide.

Author Response

The see attachment 

Reviewer 2 Report

The authors of this systematic review investigated a very important issue: how chronological age affects the chondrogenic differentiation capacity of MSCs.

The manuscript was structured according to the PRISMA guideline. Fourteen articles met the criteria for further processing. The analyses showed rather contradictory results regarding the differentiation potential of MSCs towards cartilage, bone, or adipose tissue, taking into account the age of MSC donors. They found that the differentiation capacity and potential of MSCs from older donors are inferior to those of MSCs from younger patients. It is also evident that the use and detection of markers for the identification of MSCs are not uniform.

The results of this article highlight the need to pay increased attention to the multiple, often detrimental effects of aging in the sub-classification of MSCs. The article is well written, logically structured, and analyzes a very important topic. Their results and conclusions are clear and relevant to both basic research and clinical therapy. The tables and figures are clear and easy to understand.

I recommend acceptance of the manuscript for publication.

Reviewer 3 Report

In this manuscript, Vogt et al. reviewed the roles of chronological age in mesenchymal stromal cells (MSCs). Please check the following suggestions for improving the manuscript.

[1] The information provided in this manuscript is too preliminary to be considered for publication. The manuscript only listed the study topics of some publications in the literature without summarizing what important information can be drawn from these papers. A review paper serves as guidance for a particular field; it should at least summarize the main findings in the field.

[2] The authors should provide insight into the literature and generate a unique vision for the field.

[3] The text of the manuscript does not support the title. Chronological age is not discussed as the main topic and is only discussed in the subheading section 3.2.2.

[4] The methods for literature extraction are inappropriate. How could the authors exclude 1927 duplicated studies? There are very few duplicated studies in the literature. Every publication should have at least one unique/important finding. A review article should generate insightful points based on the unique findings from each article.

[5] Too many details are provided in the tables. The entities in the tables should be as concise as possible and be those that support a particular insight of the authors of this manuscript. Entities lacking information from the literature (e.g., "Randomisation of administered dose or exposure level" and "Allocation concealment" in Table 5) should be deleted. 

[6] Based on the literature, the authors could re-organize the manuscript into sections like "Differentiation of MSCs" and "Risk factors for MSCs" and summarize/discuss the underlying mechanisms, unanswered questions/challenges, and perspectives. If the authors still plan to discuss the roles of chronological age, they could summarize and discuss what observations from the MSCs of young and older adults can be found in the literature.

[7] The English writing of the manuscript should be significantly improved. Grammar issues should be fixed in many places. Rules of scientific writing should be followed. And the manuscript should be written concisely.

The English writing of the manuscript should be significantly improved. Grammar issues should be fixed in many places. Rules of scientific writing should be followed. And the manuscript should be written concisely.

Round 2

Reviewer 1 Report

Dear authors,

Thank you for revising your manuscript taking into account the reviewers' comments which helped to improve the manuscript considerably.

However, I still found some anomalies in the insertion of references both in the text and in the final part

I ask you again to correct the citation style according to the Instructions for Authors and in the text the reference numbers must be inserted in square brackets [ ] and positioned before the punctuation; for example [1], [1–3] or [1,3].

Author Response

Dear reviewer,

Thank you for reviewing our submission. We are grateful for your comments, and the manuscript has been revised according to your comments.

Best regards,
